# Evaluation of the Relationship between Lower Limb Hypermobility and Ankle Muscle Strength in a Paediatric Population: Protocol for a Cross Sectional Study

**DOI:** 10.3390/ijerph19127264

**Published:** 2022-06-14

**Authors:** Carlos Martínez-Sebastián, Cristina Molina-García, Laura Ramos-Petersen, Gabriel Gijón-Noguerón, Angela Margaret Evans

**Affiliations:** 1Nursing and Podiatry, Universidad de Malaga Facultad de Ciencias de la Salud, 29071 Malaga, Andalucía, Spain; carlosmarseb@uma.es (C.M.-S.); gagijon@uma.es (G.G.-N.); 2Department of Podiatry, Universidad Católica San Antonio de Murcia, Campus de los Jeronimos, Guadalupe, 30107 Murcia, Spain; cmolina799@ucam.edu; 3Department of Nursing and Podiatry, University of Malaga, 29071 Malaga, Andalucía, Spain; 4Instituto de Investigación Biomedica de Malaga (IBIMA), 29010 Malaga, Andalucía, Spain; 5Discipline of Podiatry, School of Allied Health, Human Services and Sport, La Trobe University, Melbourne 3086, Australia; angela.evans@latrobe.edu.au

**Keywords:** paediatric, test, tool, strength, hypermobility, foot posture index, lunge, motor function

## Abstract

The methodological heterogeneity in paediatric foot studies does not entail a stable foundation on which to focus the diagnosis and treatment of the various childhood foot problems. For this reason, the use of highly reliable tests is essential to find relationships and to establish a basis to guide the following studies. The main objective proposed in this cross-sectional observational study protocol is to examine the relationship between hypermobility (Lower Limb Assessment Score and Beighton score) and ankle muscle strength in different types of feet. The second objective is to describe the relationship between physical activity tests in children, and to compare with foot type and ankle muscle strength. The Strengthening Reporting of Observational Studies in Epidemiology (STROBE) criteria will be followed. The hypermobility, posture, strength and physical condition tests will be analyzed through three stations, each one directed by a single specialist in paediatric podiatry. The study has been approved by the Ethics Committee of the Universidad Católica San Antonio de Murcia CE112104. The results will be disseminated regardless of the magnitude or direction of effect. Intra-examiner and inter-examiner reliability will be analyzed.

## 1. Introduction

The use of tests in the paediatric population is of great importance for assessment in diagnosis and treatment. Cochrane systematic reviews [1,2] on flat feet have confirmed that lack of evidence and clinical and methodological heterogeneity means that there are no firm conclusions on diagnosis and treatment in the child population. For this reason, many studies seek reliable measurement tools where they can find a stable base in paediatric research.

Which tests should be used in the paediatric population to assess foot and ankle function is a controversial topic, showing a lack of consensus on how foot and ankle function should be measured, defined, or assessed [3]. Therefore, high-reliability morphofunctional tests of the foot and ankle should be used [4,5,6] in order to establish an association between morphofunctional variants [7] and other variables, such as weight [8], laxity [9,10], and physical activity [11].

Previous observational studies suggested that certain specific lesions in the lower limb may correspond to different types of feet that are structurally distinguishable. A significant increased risk of stress injuries in pronated and supinated feet was associated with young soccer players [12]. By virtue of this, the clinical assessment of the alignment of the foot becomes essential for the treatment of injury and pain at the level of the lower limb.

Redmond et al. [4] described a test achieving status as an observational measurement tool to assess foot posture. The result was a six-item observational scoring system, the Foot Posture Index (FPI-6). They concluded that the FPI is a validated clinical tool that takes into account the three-dimensionality of foot posture. It has been used in children, having high reliability [13].

Other tests with high reliability in the lower limb, such as range of motion [14], ankle strength with dynamometry [5], and hypermobility tests, such as Beighton [10], or the lower limb score assessment (LLAS), have high reliability [9].

Finally, in the field of sports, physical exercise tests have been developed as a simple and inexpensive method to assess exercise capacity, which constitutes daily activities in children, promoting health. In addition, submaximal exercise tests provide a safe and practical means of assessing functional status [15].

Our hypothesis: although foot posture has been related to children’s age, more depth in morphological-functional variables of the lower limbs, e.g., hyperlaxity, strength and ankle joint range, is required. Another aspect of this research is to investigate the relationship between physical activity and morphological-skeletal variables of the lower limbs.

Therefore, the main objective proposed in the cross-sectional study protocol is to examine the relationship between hypermobility (LLAS and Beighton score) and ankle muscle strength in different types of feet. The second objective is to describe the relationship between physical activity tests in children, and to compare with foot type and ankle muscle strength.

## 2. Materials and Methods

### 2.1. Ethical Approval

This study will be carried out in accordance with the Declaration of Helsinki and it was approved by the Ethics Committee of the Universidad Católica San Antonio de Murcia CE112104.

### 2.2. Study Design

It is a cross-sectional observational study in which the Strengthening Reporting of Observational Studies in Epidemiology (STROBE) criteria will be followed.

### 2.3. Participants

In this cross-sectional study, school children aged between 5 and 10 years will participate. Measurements will be carried out during 2022. The participants will be assessed at the San Francisco de Asís school in Lorca, Murcia (Spain).

The inclusion criteria will include participants that will be aged between 5 and 10 years old, and not experiencing any foot pain at the time of the assessment. The participants will have the consent of the parents/guardians (Appendix A). Parents will be previously informed about the study via email and by signing the consent to confirm the participation of their children. Participants who have any of the following conditions will be excluded from the study: recent damage of the lower limbs, congenital structural alterations that affect distal areas of the ankle joint, as well as those cases with pathological flat feet caused by cerebral palsy, surgical treatments in the foot or lower extremity, affectations of a genetic, neurological or muscular nature.

### 2.4. Procedure

Participants and parents will receive electronic information about the nature of the study. Written informed consent will be obtained from the parents of all children, and the children’s assent will also be gained. Anthropometric data will be collected from all participants before testing. Children will be assigned a specific number to maintain confidentiality.

Hypermobility, posture, strength, and physical fitness testing will be conducted during regular physical education instruction at each school in the school’s sports hall. Participants will wear suitable shoes and light clothing (t-shirts and shorts or skirts). The children will be evaluated individually by three clinicians specialized in the paediatric field, each of whom will have been trained in the administration of all the test protocols. The data collection will involve three stations, and three examiners:Anthropometry, Foot Posture Index (FPI), Rest Calcaneal Stance Position (RCSP), Lunge Test, Beighton Scale (BS), and Lower Limb Assessment Score LLAS will be administered (Appendix A)Strength, Time to rise up from the floor (TRF), The 5 Jump-Test (5JT) will be assessed (Appendix A)Time to Walk 10 m (10MWT), Time to Run 10 m (10MRT), 6 min Walking Test (6MWT) will be assessed (Appendix A)

Each test will be explained and demonstrated before starting the test. Each test item will be performed three times, and the measurements obtained will be averaged. Participants will receive standard verbal encouragement and support throughout the tests. When a child makes a mistake during a test, the instructions and demonstrations will be repeated, and the child will be allowed to make another attempt.

According to Landis and Koch [16], coefficients of ICC that were lower than 0.20 indicated a slight agreement, 0.20–0.40 indicated fair reliability, 0.41–0.60 indicated moderate reliability, 0.61–0.80 indicated substantial reliability, and 0.81–1.00 indicated almost perfect reliability. In this protocol coefficients of ≥0.81 will be considered appropriate to consider the results of the study as valid.

#### 2.4.1. Anthropometry

Height and weight will be measured with the child dressed in light clothing and without shoes. The height of each subject will be measured with a calibrated portable SECO 7710 m. Body mass will be measured with Digital Pegasus Scales, with a margin of error of 0.05 Kg and with the subjects wearing as little clothing as appropriately possible (t-shirt and shorts or skirt) (Appendix A).

#### 2.4.2. Classification by Body Mass Index (BMI)

The BMI will be calculated using the formula BMI = weight (kg)/height (m^2^). Next, the subjects will be classified according to the cut-off lines concerning their percentile values, using the classification proposed by the F. Orbegozo Foundation in 2004 [17].

#### 2.4.3. Foot Posture Index (FPI) and Rest Calcaneal Stance Position (RCSP)

The assessment of the foot posture will be carried out by measuring the FPI with the subjects barefoot, in relaxed standing, to facilitate visual and manual inspection. The inter-examiner reliability for the FPI in the paediatric population has reached a consistent weighted Kappa value (Kw = 0.86), in a sample of children aged between 5 and 16 years [18] (Appendix A).

The RCSP will be evaluated according to the method described by Root et al. [19]. The angle between the bisector of the calcaneus and the line perpendicular to the ground will be measured. Intra-examiner reliability for RCSP in the paediatric population reached a good value of weighted Kappa (Kw = 0.61 to 0.90) [20] (Appendix A).

#### 2.4.4. Ankle Lunge Test

The range of ankle dorsiflexion will be determined by the Lunge test, which is a weight-bearing test of the range of ankle dorsiflexion when the knee is flexed. The participant will stand on a solid, horizontal surface facing a vertical wall. The trial foot will be placed perpendicular to the wall, and the contralateral foot will be placed in a comfortable and stable position. The test will involve the participant pushing the knee as far forward over the foot as possible while keeping the heel on the ground. The maximum angle of advancement of the tibia relative to the vertical will be recorded as a measure of ankle dorsiflexion using a digital inclinometer (Smart Tool™) applied to the anterior surface of the tibia (Figure 1). The intra-examiner intraclass correlation coefficients were 0.98 and the inter-examiner reliability reached an excellent value of weighted Kappa (Kw = 0.97) [6] (Appendix A).

#### 2.4.5. Beighton Scale (BS)

Beighton’s scale [10,21] will be scored for joint hypermobility at the wrist, metacarpophalangeal joint of the fifth metacarpal, elbow hyperextension, knee hyperextension (all bilateral and non-weight bearing), and lumbosacral spine [22]. The BS, when goniometry is used, is a valid instrument to measure generalized joint mobility in school-age children from 6 to 12 years of age [22]. In a study with 773 children, aged 4 to 12 years, an inter-examiner reliability for the Beighton test reached a good value of weighted Kappa (Kw = 0.81) in the pilot study [10] (Appendix A).

#### 2.4.6. Lower Limb Assessment Score (LLAS)

LLAS [7] will be assessed to measure lower limb joint hypermobility. Each limb will achieve a final score of 12 points. A score of 7/12 will indicate joint hypermobility. LLAS is reliable (inter-examiner reliability was assessed as 0.84 (ICC) [9] (Appendix A).

#### 2.4.7. Ankle Strength

Isometric muscle strength will be quantified using the Lafayette Instrument Company manual dynamometer, Model 01160, Lafayette, IN, USA. The device will be calibrated at the factory, according to the manufacturer’s data, to a sensitivity of 0.1 kg and a range of 0.0 to 199.9 kg. Each child will be placed in a seated position (hips flexed and knees extended) on an examination table with their back resting, isometric foot inversion and eversion muscle strength, and ankle plantar flexion and dorsiflexion will be measured. The sitting position will be used to view and understand the procedure, as used in this standardized protocol [23,24].

For inversion, the dynamometer will be placed distal and medial to the base of the first metatarsal; for eversion, it will be placed against the lateral base of the fifth metatarsal; for plantar flexion of the ankle, it will be placed against the metatarsal head plantar part; and for ankle dorsiflexion, it will be placed against the proximal and dorsal metatarsal heads (Figure 2).

At the beginning of the test, the reference position will be with the ankle and subtalar joints neutral [25]. The “make” method of testing will be used, in which the operator’s hand is holding the dynamometer stationary while the child exerts a maximum force against it. Previous studies confirm the highest reliability with the “make” test [26].

Three consecutive contractions of 3–5 s will be measured for each muscle group in random order. The average of three contractions is more reliable than the maximum value [27]. Additional repetitions will be performed if the evaluator feels that the child’s effort is below his or her capabilities or if the movement is wrong.

Burns et al., found the test-retest reliability for the procedure was found to be acceptable for all muscle groups. The intraclass correlation coefficient and 95% confidence interval were excellent (ICC = 0.88 to 0.95, 95% CI = 0.75 to 0.98) for all muscle groups tested. Measurement error was small for all variables (SEM = 0.3 to 0.7 kg) [28] (Appendix A).

#### 2.4.8. Simple Timed Motor Function Tests

Timed motor function tests will be explained to the children and demonstrated. All testing will be conducted with children wearing light clothing in a quiet environment below 22–26 °C. Verbal encouragement will be used during testing [11]. 

##### Time to Rise from the Floor (TRF)

The time in seconds it takes to stand from a sitting position on the ground to an upright position without assistance will be assessed [11] (Appendix A).

##### Time to Walk 10 m (10MWT)

The time in seconds for the child to walk the marked distance of 10 m will be calculated. This will be done on a surface of 20–30 m in length. The timer will be pressed to say “Go”. The stopwatch will stop when both feet cross the finish line [11] (Appendix A).

##### Time to Run 10 m (10MRT)

This will be the same as for 10MWT, but will involve the child running as fast as possible. Saying “Go” will start the stopwatch. The stopwatch will stop when both feet cross the finish line [11] (Appendix A).

##### The 5 Jump-Test (5JT)

The 5JT will be used to evaluate the explosiveness of the lower limbs [29]. The 5JT requires an elongated flat space of 15 m. According to Chamari et al., it consists of 5 consecutive strides with the position of the feet joined at the beginning and the end of the jumps. From the initial position, the participant will have to jump directly forward with one leg and after the first 4 strides, that is, alternating the left and right foot 2 times each, they will perform the last stride and finish the test with both feet together. The best performance (greatest distance) will be recorded (Appendix A).

##### 6 min Walking Test (6MWT)

The 6MWT will be performed on a flat surface in a 30 m long covered corridor marked every 2 m, where the child will walk for 6 min, being able to rest while performing the test. The test will be performed following the recommendations of the American Thoracic Society [30] (Appendix A).

### 2.5. Sample Size

The sample size will be determined by application of the EPIDAT program (https://www. sergas.es/Saude-publica/EPIDAT?idioma=es). The study will be designed to detect changes exceeding 0.8 (high effect size) with a type I error of 0.05 and a type II error of 0.2. This is based on prior recommendation of 15 participants for pilot studies to estimate outcome variance and allows for a predicted attrition rate of 20% with a precision of ±5% with 95% confidence level [31].

### 2.6. Data Management and Analysis

Quantitative data will be evaluated using SPSS (IBM SPSS Statistics: V.24, USA). The results obtained will be reported as a median and interquartile range, if the distribution of the variables is not normal, and as mean and standard deviation (SD), when there is normal distribution. The normality of the distributions will be examined using the Kolmogorov-Smirnov test and the intra-examiner reliability of the measurement instruments will be calculated using a two-way mixed consistency ICC model. The bivariate analysis will be performed with Student’s *t*-test and Wilcoxon’s non-parametric test. The level of significance will be set at *p* < 0.05.

## Figures and Tables

**Figure 1 ijerph-19-07264-f001:**
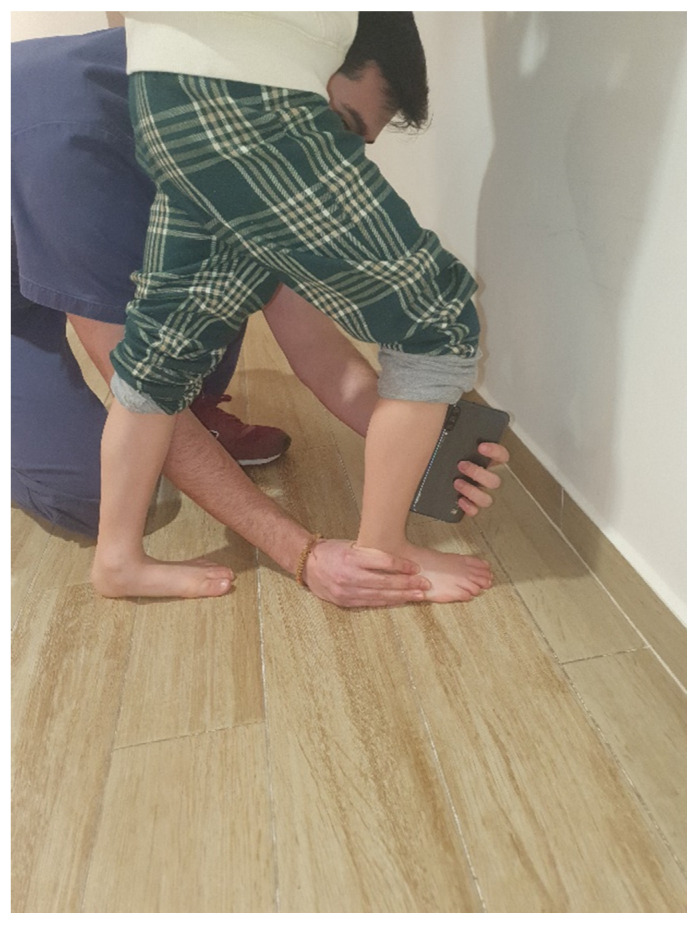
Assessment of range of ankle dorsiflexion by the Ankle Lunge Test.

**Figure 2 ijerph-19-07264-f002:**
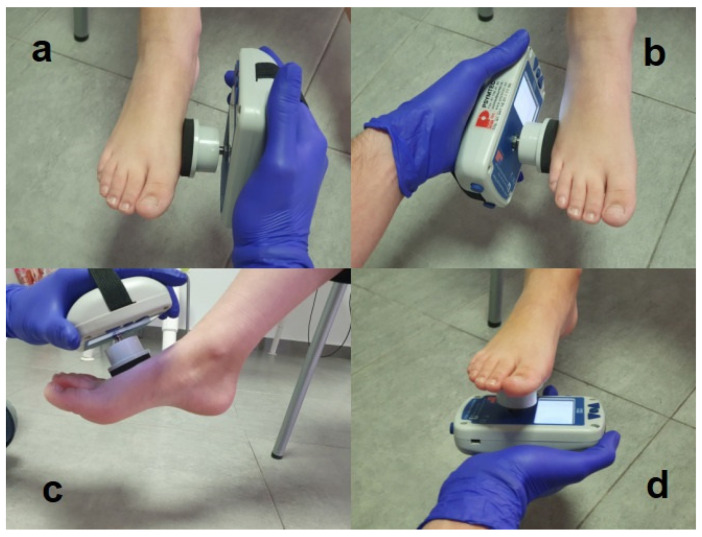
Assessment of ankle strength using the dynamometer: (**a**) inversion assessment, (**b**) eversion assessment, (**c**) dorsiflexion assessment, (**d**) plantar flexion assessment.

## Data Availability

The data presented in this study are available on request from the corresponding author. The data are not publicly available due to privacy.

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
