# Peer review of "Evaluation of the Relationship between Lower Limb Hypermobility and Ankle Muscle Strength in a Paediatric Population: Protocol for a Cross Sectional Study"

_ijerph, 2022, doi:10.3390/ijerph19127264_

Round 1

Reviewer 1 Report

  The authors made the suggested changes and rebuilt the article significantly. I still have concerns whether such a study protocol is needed and necessary to assess lower limb hypermobility and ankle muscle strength in a pediatric population. The examination is long and I do not know if it will bring anything new to orthopedists. What is the practical application of this test protocol? Is this test necessary?

Author Response

Thank you very much for giving us the possibility of addressing all the questions that arose during the review process. We think the reviewer’s comments have greatly improved the quality of our study protocol

Reviewer 2 Report

The authors are describing a protocol, but does not appear to have collected any actual data or developed any results or conclusions from the protocol.  I would encourage the authors to resubmit with results/discussion/conclusion.

Author Response

(The authors gave the same response as above.)

Round 2

Reviewer 1 Report

The authors explained the idea behind the study. Manuscript for approval.

This manuscript is a resubmission of an earlier submission. The following is a list of the peer review reports and author responses from that submission.

Round 1

Reviewer 1 Report

The authors report that certain specific lesions in the lower limb may correspond to different types of foot that are structurally distinguishable, but do not indicate which. The authors formulate a research protocol to evaluation of the relationship between lower limb hypermobility and ankle muscle strength in children, but they do not explain why the indicated tests were selected, what were the test selection criteria, whether the tests were analyzed that were rejected as unsuitable for problem assessment (eg. what minimum value of weighted Kappa was accepted).  

Some questions and remarks:
The article, which aims to identify a universal cross-sectional research protocol, was based on only six articles from the last ten years. The bibliometric data (scholar google only) shows over 2,000 articles dealing with lower limb hypermobility in paediatrics since 2012. I consider the introduction to be poor. The authors indicate, for example, that only young footballers were exposed to foot injuries (Ref.3).

How was the stability of the operator's hand determined in the tests of ankle strength, was the situation in which the operator presses the device assessed?

Why was the protocol verification not presented in own research? There are no results, therefore, the protocol cannot be assessed for operator independence on test results and the relationship between hypermobility and ankle muscle strength in different types of feet, as well as the relationship between physical activity tests in children, and to compare with type of foot and ankle muscle strength.

There should be more diagrams or photos in the article, I think they would enrich the protocol.

The EPIDAT program requires more details.

The appendices are too extensive and carelessly prepared.

I suggest not to use abbreviations as chapter titles.

I suggest you improve the selection of literature. References 11 and 27 require more detail.

The article contains some writing mistakes (lines: 115, 119, 131, 149, 176, 193, Appendix B: in tables).

Reviewer 2 Report

Interesting topic, but this is not an article but a research protocol.

It cannot be published in this form.

An article containing the results of the study can be published.

Lots of stylistic, grammatical and punctuation errors.

The title reads "...... lower limb hypermobility ...." And the article is about the foot. Please change the title.

Abstract:

Please delete the bioethics committee consent information.

Research on the basis of this protocol should be performed, the results analyzed, and only then can the article be published.

You have to create a discussion and conclusions based on the research results - this is missing now.